# Solutions to a Radical Problem: Overview of Current and Future Treatment Strategies in Leber’s Hereditary Opic Neuropathy

**DOI:** 10.3390/ijms232113205

**Published:** 2022-10-30

**Authors:** Samuel J. Spiegel, Alfredo A. Sadun

**Affiliations:** 1Gavin Herbert Eye Institute, University of California, Irvine, CA 92617, USA; 2Jules Stein and Doheny Eye Institute, University of California, Los Angeles, CA 90095, USA

**Keywords:** leber’s hereditary optic neuropathy, optic neuropathy, mitochondrial disorder, mitochondrial optic neuropathy, hereditary optic neuropathy, neuro-ophthalmology, idebenone

## Abstract

Leber’s Hereditary Optic Neuropathy (LHON) is the most common primary mitochondrial DNA disorder. It is characterized by bilateral severe central subacute vision loss due to specific loss of Retinal Ganglion Cells and their axons. Historically, treatment options have been quite limited, but ongoing clinical trials show promise, with significant advances being made in the testing of free radical scavengers and gene therapy. In this review, we summarize management strategies and rational of treatment based on current insights from molecular research. This includes preventative recommendations for unaffected genetic carriers, current medical and supportive treatments for those affected, and emerging evidence for future potential therapeutics.

## 1. Introduction

Leber’s Hereditary optic neuropathy (LHON) is one of the inherited hereditary mitochondrial optic neuropathies. Inherited optic neuropathies have been estimated to affect 1 in 10,000 individuals and are an important cause of visual impairment [1]. LHON has been reported to be the most common primary mitochondrial DNA disorder with epidemiological estimates ranging from 1 in 27,000 to 40,000 [2]. Clinically, it primarily presents in individuals in the second or third decade of life. Typically, these young adults experience unilateral, painless, subacute central or cecocentral scotomas, impaired color vision, and visual acuity loss with involvement of the contralateral eye in the following weeks to month. It is a clinical diagnosis which is subsequently confirmed by blood testing for mitochondrial DNA (mtDNA) analysis. The 3 most common mtDNA mutations (*m.3460G>A*, *m.11778G>A*, and *m.14484T*>C mutations) account for 90–95% of cases [3,4,5]. Genetics (mitochondrial and likely nuclear) and environmental factors may lend to respiratory chain dysfunction leading to retinal ganglion cell (RCG) dysfunction, cell death and vision loss. Currently most persons affected will have irreversible severe visual impairment.

To date, treatments have been limited and there is no curative therapy. The European union has authorized the use of Idebenone in patients with LHON and current consensus guidelines recommend its use in affected individuals [2]. Although efficacious treatment options are limited for those affected by LHON, recent years have seen an upsurge of scientific advancements in the understanding of the disease process and search for potential therapeutics.

LHON is uniquely placed for scientific study as most cases occur from identifiable point mutations, it affects a unique and measurable cellular line, and is one of the more commonly inherited mitochondrial disorders. Clinicians and scientists have been working on further defining the clinical disease course of LHON in combination with the molecular impact of genetic and environmental factors. This review will discuss the natural disease course and molecular mechanisms in the development of LHON to allow for an in-depth review of the current and future treatment strategies in LHON. It is this multifaceted understanding of specific pathways of damage and their influence on LHON and visual outcomes which have provided insights to allow for the creation of novel therapeutic strategies. Additionally, as LHON has become a model for mitochondrial disorders, the advancements in this field serve to inform research of other related disorders.

## 2. Natural Disease Course

LHON has been subcategorized into disease states to help further define and approach clinical interventions (Table 1). It is important to understand these key time markers within the disease process for the clinical and therapeutic implications. Broadly individuals with LHON mtDNA mutations can be classified into carriers and affected symptomatic patients.

Currently, the asymptomatic phase is defined as any individual who is a carrier of one of the known causative mutations but is not experiencing visual loss. Due to highly variable, incomplete penetrance, LHON individuals may never develop vision loss. Carriers and pre-symptomatic patients may demonstrate clinical findings in the absence of subjective vision changes. Inferotemporal retinal nerve fiber layer (RNFL) swelling can be seen on fundoscopic exam or Optical Coherence Tomography (OCT) and is not indicative of progression to vision loss [6,7]. Pre-symptomatic patients additionally may have subtle clinical findings such as mild dyschromatopsia, reduced contrast sensitivity, and ultimately will show fundus changes such as telangiectatic vessels. More recent studies have also demonstrated subnormal electroretinogram and visual evoked potentials in these patients [6]. OCT angiography studies have shown that microvascular changes occur in the temporal sector and may precede RNFL changes and mirror the ganglion cell layer changes [8,9,10]. The optic disc microangiopathy, seen as telangiectatic vessels, may occur in both asymptomatic and acute stages of the disease. Its role in the pathogenesis of LHON is currently not well understood [8]. Pre-clinical or pre-symptomatic individuals may eventually go on to develop vision loss. Some may define conversion to the affected stage when loss of macular RGCs occur while visual acuity is still normal; however, most define conversion when the onset of vision loss occurs. 

Those who convert from asymptomatic to symptomatic states typically do so in their second or third decade of life [11]. Disease onset is characterized by subacute, central vision loss occurring sequentially in the contralateral eye within weeks to months of the first eye in >97% of individuals [12,13]. Visual function will decline in the following months with significant impairments in visual acuity and color vision. Visual fields demonstrate dense central or cecocentral scotomas and OCT will show loss of macular retinal ganglion cells (RGCs). The evolution and progression of disease occurs during the acute/subacute phase. Most individuals will continue to lose visual acuity during the first six months, at which point central vision loss may stabilize, but quantitative clinical metrics may continue to decline (for example visual field loss and OCT measurements). Why vision loss is catastrophic and not more gradual is still not well understood. Evidence is limited given the lack of longitudinal studies in pre-clinical patients. Currently mechanical and metabolic mechanisms are hypothesized in the development of conversion to symptomatically affected patient. Metabolic changes within nerve fibers lead to increased mitochondrial and axonal stasis, eventually producing axonal swelling. Mechanically, the laminar portion of the optic nerve is anatomically unable to accommodate swelling, thereby leading to vascular insufficiency and triggering catastrophic retinal ganglion cell loss [8,14,15,16]. It is the papillomacular bundle (PMB) which is selectively lost early in the disease course and this pathologic process affects the smallest fibers first [17]. Further understanding of changes seen from carrier to pre-symptomatic to symptomatic is needed to better understand disease conversion.

After about 1 year stability occurs, and it is at this time patients are considered to have transitioned to the chronic phase of the disease [18,19]. Some individuals will show improvement between 18–24 months after onset of disease. There may be reduction of the central scotoma or in many cases a fenestration of the central scotoma may lead to an opening of a central visual island allowing improved central acuity. Recovery is currently most dependent on mutation subtype and age at onset. The best potential for recovery is seen in those younger than age 12 at onset and those with the m.14484T>A variant [20]. Additionally subacute presentation and large optic discs predispose to better recovery. Adults with the *m.11778G>A* mutation have been found to have the worst visual outcomes [13,21]. Most patient remain severely affected and rarely are visual acuities better than 20/200. 

It should be noted that the disease state categories above are applied to help define disease process for clinical and scientific trials; however, it is becoming further elucidated that the LHON phenotype is fairly heterogenous. There is variability between each mutation in regard to timing, severity and outcomes. Additionally, within specific subgroups of genetic mutations there is variability given environmental and genetic heterogeneity within the population, for example mtDNA haplogroup polymorphisms [22].

There appears to be two important subcategories of clinical manifestation that has been termed LHON Type I and Type II. Type I is likely to be a mtDNA mutation in the context of nDNA which is susceptible. This gene-gene process may be inevitable. Type II may be Type I cases in which environmental factors are key and lead to conversion. This might explain several distinctions in presentation. Type I is abrupt (sub-acute), and Type II is more insidious. Type I occurs about age 20 and Type II often after age 40. Type II is almost always associated with a major exposure to smoke or smoking. Type II is more likely to recover if the patients risk factor is removed. Most importantly, in Type I there is profound loss of both structure (OCT-RNFL) and function (VFs), but in Type II there is often a structure-function mismatch (RNFL is partly preserved) [12]. 

## 3. Molecular Background and Pathophysiology

Mitochondrial respiration is driven by redox reactions, organized through mitochondrial electron transport chain complexes. The complex I (a NADH:Ubiquinone oxidoreductase) proton pump is the first step in the respiratory chain and couples electron transfer with proton translocation across the mitochondrial membrane. This complex contains 45 subunits, 7 of which are mtDNA encoded [23]. The 3 most common mtDNA mutations affect mitochondrial Complex I at the ND1 (*m.3460G>A*), ND4 (*m.11778G>A*), and ND6 (*m.14484T>C*) subunits. Complex I then uses two co-factors: coenzyme Q (ubiquinone, CoQ) and Cytochrome c (cyt c) to transfer electrons to the succeeding respiratory complexes (Figure 1). LHON mutation *m.3460G>A* has been demonstrated to alter the catalytic activity of Complex I with subsequent impairment in mitochondrial respiration and all three mutations impair mitochondrial respiration and reduce complex I-drive ATP synthesis. Of particular importance, the electrons produce reactive oxygen species (ROS). There is also, to a lesser extent, a reduction in ATP synthesis. Figure 2 shows why the retinal ganglion cell (RGC), with a long non-myelinated retinal nerve fiber layer (RNFL) is particularly vulnerable to this process. The generation of ROS occur when electrons spill from the transfer between Complex I and CoEnzymeQ10 and react with molecular oxygen, triggering a cascade of deleterious downstream reactions and eventual apoptosis [24]. Most of the energy required by neurons is to re-establish the membrane potential after an action potential. Fortunately, this only has to happen at the Nodes of Ranvier as myelin covers the remaining membrane. However, RGCs being unmyelinated in the eye, have anatomical and biochemical elements that predispose them for failure. Since RCGs contain the bioenergetically demanding unmyelinated RNFL, they have a high concentration of mitochondria most concentrated in the prelaminar and intralaminar portions (Figure 2). This predisposes them to mitochondrial dysfunction [24,25]. Specifically, due to the adverse surface to volume characteristics, the thin axons of the papillomacular bundle and their corresponding RGCs are most vulnerable and are selectively lost early in LHON (Figure 3). It is this selective loss that has allowed researchers to further delve into pathophysiologic biomolecular mechanisms by which mitochondrial impairment creates the devastating cascade of apoptosis. From this research various therapeutic models have been attempted and are being further investigated. 

## 4. Unaffected Carriers and Pre-Symptomatic/Pre-Clinical

Currently, there is no recommended treatment for those individuals who are known carriers of the LHON mutation [2]. That being said, there are important factors to consider in these individuals based on the newest insights we have available regarding LHON. Genetic counseling and lifestyle modification are the current focus of neuro-ophthalmologic consensus statements, but other factors are under investigation for potentially mitigating risk of vision loss. 

### 4.1. Genetics and Counselling

As LHON is a maternally inherited, a male (affected or unaffected) with a known pathogenic mtDNA mutation cannot transmit this to any of his offspring, and a female with a similar mutation will transmit it to all her offspring. As LHON exhibits incomplete penetrance the appearance of disease may vary widely within each family or between families with the same mtDNA mutation. 60% of families will have a history of visual loss affecting maternal relatives. Incomplete penetrance further complicates genetic counseling because it is unknown when or if an LHON mutation carrier will become symptomatic. Unaffected family members should be cautioned that finding the mutation in relatives in maternal line is very likely, and the results will not aide in prognosis. The current international consensus statement released in 2021 is that all maternally related family member do not currently need to be tested. It is generally agreed upon however that they should be screened [2].

If individuals do receive genetic testing, the large majority of positive individuals will be homoplasmic. Heteroplasmy is well documented and genetic testing results documenting heteroplasmy must be carefully interpreted as mtDNA in peripheral blood may not reflect the heteroplasmic load in the RGCs [22]. Along similar lines, preimplantation or prenatal testing for a pregnancy may be done, but there is variability in the mtDNA in amniocyte and chorionic villi and results may not correspond to levels found in RGCs [22]. Therefore, its interpretation should be done cautiously and by experienced providers. In addition, because of non-life-threatening nature of LHON and its highly variable penetrance there are clinical and ethical considerations on if preimplantation or prenatal testing is warranted. Newer genetic developments propose alternative future potential options for genetic modifications. For example, mitochondrial donation from a donated egg or zygote with healthy mitochondria or removing the nucleus and replacing it with the nucleus from the egg or zygote from the affected mother has been done, but again this is a developing field with many ethical considerations [27,28].

Lastly, haplotype J is now well documented as leading to a higher conversion and therefore increases the penetrance of LHON mutations. The phenomena is felt to be due to lower mtDNA and mtDNA-encoded polypeptides This is currently of less importance in regard to genetic counselling clinically, but if known for a carrier may lead to some prognostic value [22,29]. 

### 4.2. Neurotoxins

The incomplete penetrance seen in LHON lends to the classic hypothesis and well accepted statement that mtDNA mutations are necessary for an individual to be affected but alone are insufficient to cause vision loss. It is the addition of environmental factors which allows for conversion from asymptomatic individuals (pre-clinical) to affected individuals (clinical). A growing body of data is now available on mitochondrial toxicity, in particular increases in oxidative stress induced by tobacco smoke, other smoke, and alcohol.

Multiple studies have shown that alcohol consumption at high levels, especially binge-drinking, and smoking tobacco have a strong association with more severe symptoms and prognosis in LHON carriers [12,30]. Affected smokers have demonstrated a lower mtDNA copy number as compared with affected non-smokers and with unaffected mutation carriers. Tobacco smoking directly affects the compensatory mechanism counteracting the pathogenic effects of LHON mutations. This regulation of mitochondrial biogenesis and mtDNA copy number has been proven to be crucial in development of vision loss in LHON, having a direct impact on the disease penetrance [31]. These molecular effects have been demonstrated in clinical patients as well. Subgroups of LHON patients who present with delayed onset are found to have increased use of tobacco products. It is hypothesized that these patients are a unique subset that may have been unaffected carriers and became affected later in life after many years of smoking [12]. More recently during the recent COVID-19 pandemic, significant increase in substance abuse, specifically EtOH intake and cigarette smoke, has led to reported conversion of LHON carrier to affected individuals. One case series recently published reported 3 LHON patients all of which were atypical onset > 50 yrs old who had dramatically increased their exposure to EtOH during the COVID 19 pandemic [32]. It is more than just cigarette smoke but exposure to smoke in general that is felt to be associated with disease conversion. There are case reported instances of individuals becoming affected after other forms of smoke inhalation, such as a woodburning stove or rubber tire fires. (Sanchez, JNO 2006) Therefore, it is recommended to not only avoid tobacco related smoke but all smoke inhalations. 

### 4.3. Mitochondrial Biogenesis, Autophagy, and Transmitophagy

Increasing in the mitochondrial quantity and mtDNA copy number to increase total ATP production is a theorized method in which one may prevent disease conversion in pre-symptomatic individuals. This compensatory mechanism has been highlighted in LHON and it has been postulated that increased mitochondrial biogenesis may be a responsible driving factor in its incomplete penetrance [33]. Although there is currently little known in regard to clinical effectiveness in this field it poses an extremely interesting and targetable mechanism of therapeutic intervention. Systemic mitobiogenesis, occurring mainly near the cellular soma is currently being further elucidated. Optic nerve and RCGs pose further challenges given the length of their axons and distance mitochondria would need to be transported. Local mitochondrial biogenesis is likely important in LHON but is currently not well defined in RCGs [34]. Specifically, a quantitative increase of the mitochondrial mass has been attempted to overcome mitochondrial functional deficiencies [34,35]. 

Additionally, dysfunctional mitochondria are implicated in the triggering of apoptosis. The quality control of mitochondria requires a complex balance of mitobiogenesis and autophagy. Autophagy involves lysosomal degradation and elimination of dysfunctional and damaged proteins and organelles. Autophagy may be significantly compromised in cells with LHON mtDNA mutations. This results in reduced clearance of dysfunctional mitochondria (mitophagy) and is a contributing factor to cellular impairment and death. Pharmacologic activation of mitophagy in LHON cell models has been shown to selectively clear damaged mitochondria and improves overall cellular survival [36]. Transmitophagy is a process unique to RGCs in whichdamaged mitochondria in RCGs coalesce and are exocytosed from the myelinated portion of these axons. They are taken up by surrounding astrocytes and degraded. This helps regulate the mitochondrial number and quality. It is this quality control mitophagy that may become a new therapeutic approach that restores the balance of functional and dysfunctional mitochondria [36]. 

Many drugs have been studied, among them metformin, rosiglitazone, resveratrol, vitamin B3 and NAD+ precursors which have shown promising results; but no studies specifically relate to vision loss or optic nerve tissues [37,38,39]. Sets of drugs has been associated with an increased mtDNA amount. Lipoic acid, lipamide, thiazolidinediones and polyphenols have been shown to increase mtDNA copy number in animal models which can be associated with increased mitochondrial proteins, quantity, activity of OXPHOS complexes and ATP amount [34,40]. By targeting these three mechanisms: mtDNA copy, mitobiogenesis, and transmitophagy we may be able to create a new cellular homeostasis that is less predisposed to cellular damage and eventual vision loss in patients with LHON mutations. 

### 4.4. Estrogens

In relation to the section above, estrogens should be discussed during the pre-symptomatic stage of LHON, particularly when it comes to prevention in female patients, as this has one of the largest bodies of literature in relation to potential neuroprotection in LHON. Reduced prevalence of LHON affected women is well established, but there is also an increased incidence of disease seen with patients that may correlate with a decline in estrogens (menopausal patients). It has been found that estradiol levels increase mtDNA content, oxygen consumption and ATP levels in human cybrids with LHON mutations [41]. In vitro studies have shown that specific estrogen receptors localize to the mitochondrial networks of RGCs. Through activation of estrogen receptors, there is increased antioxidant enzyme production (e.g., Superoxide dismutase 2), promotion of mitochondrial biogenesis, and increased mtDNA copy. This in turn enhances cellular viability, ameliorating ROS production and reducing the rate of apoptosis [42]. Although not currently standard of practice there is discussion on the benefits of estrogen therapy in patient who are known unaffected carriers and nearing the age of menopausal onset and is currently an evolving area within the field. 

## 5. Symptomatic Phase

### 5.1. Antioxidants

Oxidative stress and its modulation is a focal point in mitochondrial disease. The potential benefit in using antioxidants to help reduce the neurotoxic strain imposed on RGCs by ROS is quite a desirable theoretical avenue of treatment. The hypothesized mechanism of reducing the neurotoxic stress from multiple supplements (e.g., vitamins B2, 3, 9, 12, C, ubiquinone, carnitine, L arginine, alpha-lipoic acid amongst others) and their combinations have been attempted to help reduce this ROS imposed stress. None so far have been shown to have any significant benefit in LHON and currently there is not enough scientific evidence to support their use clinically. 

### 5.2. Coenzyme Q_10_ (Coq10)

Of the various over the counter options, the ubiquinone family seemed the most promising for effective treatment. This coenzyme shuttles electrons between complex I and II to complex III of the ETC. The reduced CoQ10 therefore also has antioxidant properties, minimizing ROS generation. Additionally, CoQ10 might deliver the electron as needed to complex III thus restoring energy production. However, the major hindrance to its effect is its inability to cross cell membranes, the mitochondrial membrane, and blood–brain barrier due to its long lipophilic tail and therefore low bioavailability. It has been shown to be partly effective in other mitochondrial related disorders but has not panned out for LHON [43,44]. In order to overcome the issues of drug delivery with CoQ10, Idebenone was developed.

### 5.3. Idebenone

Idebenone is a synthetically derived molecule which similarly facilitates electron transfer between ETC complexes allowing for improved ATP production and minimized ROS generation [45]. This soluble analog was first shown to be protective in disorders affecting complex I (such as LHON) in both animal and human subjects. Idebenone is activated in the cytoplasm to its oxidized form by NAD(P)H:quinone oxidoreductase (NQO1) thus allowing for conversion into its reduced form to shuttle electrons directly to complex III, bypassing the complex I dysfunction seen in LHON. The formation of Idebenone to its oxidized form therefore appears essential to its therapeutic efficacy, but this oxidized form also has been shown to have potential inhibitory and adverse effects on complex I. Variable expression of NQO1has been postulated to contribute to cases in which Idebenone is ineffective [46,47,48]. Preclinical studies did confirm a cell specific increase in ATP production and reduced ROS levels in fibroblasts of LHON patients and prevention of RGC loss in LHON mouse models [49,50]. Since the first human case reports in 1992 it has developed a tremendous body of literature. 

In 2011 a randomized clinical trial “Rescue of Hereditary Optic Disease Outpatient Study” (RHODOS) presented evidence that idebenone can be beneficial to preserve and/or restore vision [51]. This prospective, placebo-controlled trial evaluated patients with <5 years of visual loss and a trend was noticed in favor of idebenone when the authors analyzed changes in best visual acuity and excluded patients with the 14484 mutation (higher rate of spontaneous recovery). Patients with more recent onset of vision loss were more likely to demonstrate improvement. Shortly after this, Carelli et al. published a retrospective study showing that the proportion of patients treated with idebenone within 1 year after visual loss in the second eye at varying doses showed demonstratable improvement compared to untreated patients [52]. Additionally, earlier visual improvement was associated with prompt start and longer duration of therapy [52]. 

These results then triggered a seminal event in LHON treatment, the European Union’s authorization of the use of Idebenone in patients with LHON in 2015. This was then followed by the 2017 consensus conference to address therapeutic issues in the treatment of LHON. The consensus statement and most current recommendation is that Idebenone should be started as soon as possible in patients with disease onset of less than 1 year. Additionally, in these subacute/dynamic patients, treatment should be continued for at least 1 year to assess the start of therapeutic response or until a plateau in terms of improvement is reached [2]. With this medication available, the therapeutic window should not be lost by a delay of diagnosis [52,53].

Since then, the previously reported beneficial effect of idebenone on recovery and preservation of vision has been confirmed by multiple other major studies, including the expanded access program (EAP) [54], a Japanese prospective, interventional study [55], and a Netherlands national cohort study [56], and the soon to be published LEROS study—(NCT02774005).

### 5.4. Elampritide

Elamipretide (MTP-131) is an antioxidant peptide which has been found to increase ATP synthesis as well as reduce ROS production [57]. It targets cardiolipin selectively within the inner membrane of mitochondria and prevents conversion of cytochrome c into peroxidase. Sadun et al. conducted a Phase II Clinical Study (ReSIGHT) of which the results are not yet published but results have been updated as of November 2021—(NCT02693119). In total 12 LHON patients were recruited and treated for 52 weeks. No difference in BCVA was observed, but because of a trend towards improvement, all 12 patients completed an open-label extension with bilateral treatment for a total of at least 84 weeks [47]. 

### 5.5. EPI-743

Alpha-tocotrienol quinone (EPI-734) is a third generation quinone molecule that has been studied in depth in vitro and needs further investigation to determine benefit in LHON patients. It has been used and studied in other inherited mitochondrial diseases and has been found to be approximately 1000 to 10,000 fold more potent than coenzyme Q10 (because of drug delivery) or idebenone (because of electron delivery) in protecting mitochondria. It is an antioxidant para-benzoquinone that also replenishes glutathione, acting on oxidoreductase enzymes. Case reports or case series support its benefit in LHON; the largest to date being a small open-label trial, in which EPI-743 arrested disease progression and reversed vision loss in 4 of 5 treated patients with LHON [58,59].

### 5.6. Cyclosporine

Drugs or molecules developed for use in other pathologic processes which play a role in the modulation of the cellar intrinsic pathway or apoptosis may lead to promising therapy. Cyclosporine A inhibits opening of the mitochondrial permeability transition pore and is one of the anti-apoptotic drugs that has been studied in LHON. Oral cyclosporine (2.5 mg/kg/day) was given to patients in the hope of preventing second-eye involvement in patients with strictly unilateral Leber’s hereditary optic neuropathy. Despite cyclosporine treatment, second-eye involvement occurred in all five patients included in the study, resulting in severe loss of vision, down to 20/200 or less. In addition, there was also a worsening of the visual acuity, the mean visual field defect, and the average thickness of the GC-IPL in the first eye affected [60]. Other molecules targeting different signals along the apoptotic pathways remain good candidates for future consideration.

### 5.7. Light Therapy and Electrical Stimulation

Photons from Near Infrared Light (NIR) light can penetrate diseased retina, be absorbed by mitochondrial photoreceptors, such as cytochrome c oxidase, and potentially promote mitochondrial energy production. Evidence has suggested that NIR light may restore biological function of damaged mitochondria, upregulate cytoprotective factors and inhibit apoptosis [61]. NIR light can also inhibit cell degeneration of retinal ganglion cells caused by inhibitor rotenone to the mitochondrial complex [62,63]. This has been postulated as a potential therapeutic option in LHON. Others, however, are concerns that upregulating OXPHOS with NIR may, by increasing mitochondrial metabolism also increase ROS, worsening the clinic outcome in LHON. The FDA approved the usage of this in patients and a clinical trial was initiated; however, it failed to recruit enough patients. An unpublished study in the Brazilian pedigree failed to show any benefit from NIR.

Electrical stimulation techniques may have shown potential benefits for treatment in diseases which affect the retina and optic nerve. Non-invasive electrical stimulation has been evaluated in traumatic non-arteritis optic neuropathies as well as retinal disease [64]. Recent small preliminary studies have looked at safety and efficacy of skin electrical stimulation in patients with chronic phase LHON with some promising results. Current studies have a lack of randomization and/or control arm, therefore further research is needed [65,66].

### 5.8. Gene Therapy

The current therapies discussed thus have focused on prevention and compensation. These pharmacologic mechanisms have been ones which ameliorate mitochondrial dysfunction at the cellular level by reducing, or attempting to reduce, ROS generation and the impact of mtDNA mutations on cellular homeostasis. Gene therapy, which alternatively targets replacement and repair of dysfunctional or damaged cellular pathways, poses an exciting new approach in the field. 

In order to deliver a gene product to a mitochondrion there are several steps. The vector must endocytose within the cell, internalize within the mitochondria, and affect mitochondrial metabolism. This is very problematic, given that many DNA molecules do not cross mitochondrial membranes unaided. Additionally, mitochondrial genes require allotopic expression. Allotopic expression of mitochondrial genes is the deliberate relocation of mitochondrial gene into the nucleus followed by the importation of the genetically encoded polypeptide from the cytoplasm into mitochondria. The most studied mechanism to date for replacement of dysfunctional mtDNA is via intravitreal (IVT) injection of viral vectors. 

The currently most established gene therapy consists of a wild-type of ND4 subunit packaged into an adeno-associated-virus 2 (AAV2) which is injected into the vitreous and targets the closest cells which are RGCs near the macula. In theory, upon cellular transfection, the wild-type ND4 gene is allotopically expressed. The AAV2 gene therapy vector carrying the wild-type ND4 gene gets transported to the nucleus where it is transcribed into messenger RNA which is later transcribed by ribosomes. The genetic sequences encoded also optimize translocation into the inner mitochondrial matrix and allow its integration within complex I to restore function [67]. Preclinical studies have demonstrated that recombinant AAV2 with wild type ND4 can rescue ATP production in cultured fibroblasts isolated from ND4-LHON patients, and that the therapeutic ND4 protein could successfully integrate into complex I in induced LHON models, preventing RGC apoptosis and optic nerve atrophy [68].

To date, there have been several phase I, II and III studies for LHON gene therapy. RESCUE and REVERSE are two randomized, double-blind, sham-controlled, multi-center, phase III clinical studies in which an intravitreal injection administered AAV recombinant wild type ND4 in one eye and an intravitreal administration of sham injection was delivered in the fellow eye [69]. In 2017, these phase III clinical trials studied the effects of single eye injection with GS010, a recombinant, AAV, containing a modified cDNA encoding the human wildtype ND4. They only differed in the duration of vision loss (≤6 months for RESCUE, and >6 months to 1 year for REVERSE). 37 patients with visual acuity loss were included in the RESCUE study and 39 patients were included in the REVERSE study. On average, patients experienced an improvement in their visual acuity of about three lines. The surprising outcome from these trials was that a similar improvement occurred in the contralateral (sham treated) eye [70]. A non-clinical trial on primates suggested a possible retrograde, trans-chiasmatic transit of the vector from the treated eye to the sham-treated eye as an explanation for the unexpected bilateral visual improvement found in the RESCUE and REVERSE studies; however, this is a hotly debated topic in the neuro-ophthalmologic community. Some experts doubt that the number of virions that would make it to the contralateral eye is sufficient to infect enough RCGs for a clinical benefit. In 2019, the enrollment of 98 patients in a new Phase III clinical trial (REFLECT) was completed. The REFLECT study evaluated the efficacy and safety of bilateral intravitreal injections in subjects with 11,778 LHON mutation and follow up has shown that the statistically significant improvement of BCVA from baseline and the nadir reported at 1.5 years post administration was maintained at 2 years [71]. The improvement observed in placebo-treated eyes is consistent with the contralateral effect of a unilateral injection which was previously reported in RESCUE and REVERSE [71]. 

## 6. Chronic Phase

Currently there is not enough evidence to recommend treatment in patients in the later chronic stage who have experienced vision loss bilaterally, and there is no evidence to recommend treatment beyond 5 years from onset [2]. Overall, treatment in these individuals remains mostly supportive. Although there is no treatment that can stimulate optic nerve regeneration and repair, prevention of further visual loss by mitigation of risk factors, and supportive assistance for the patients, has great value.

### 6.1. Supportive Therapy

Currently, with limited available treatment options, the majority of patients with symptomatic LHON will progress to legal blindness (best corrected visual acuity of less than 20/200). As previously mentioned, many will have central scotomas with residual peripheral vision and are good candidates for low vision therapy and rehabilitation services. Additionally, those patients with a fenestration through which they have improved acuity, may be able to maximize this through assistive devices to maintain independence. Ample digital devises, such as smart phones or iPads, can be very helpful. Evidence supporting low vision rehabilitation is less robust, and a recent 2020 Cochrane review found that there was low-certainty evidence that some rehabilitation interventions improved vision related quality of life compared to usual care. Particularly, psychological therapy and methods of enhancing vision had the best evidence [72]. Overall, therapy should consist of a multifaceted approach including methods of enhancing vision, such as magnifying devices, electronic devices, or other technologies to improve remaining vision in combination with multidisciplinary rehabilitation programs such as balance training, occupational therapy, and home safety assessments. Most recently, qualitative data from LHON focus group interviews found that patients were hopeful that therapy would restore autonomy and improve their ability to enjoy a fulfilling life, while alleviating financial demands and demands placed on relatives [73]. 

### 6.2. Psychological Counseling

The impact of LHON extends beyond visual and activity related limitations. An important aspect of care for patients with LHON is the psychologic impact it has on both patients and carriers [74,75]. As LHON exhibits incomplete penetrance there may be multiple siblings or family members which are at various disease categories—carrier, pre-symptomatic, or symptomatic. It is important to be sensitive to the interfamilial dynamics and aware of the anxiety it may provoke. Vision-related quality of life is significantly reduced in LHON patients who also have higher levels of depressive symptoms compared to unaffected carriers [74,76]. LHON patients are at increased risk of anxiety and depression. Vision-related quality of life and depressive symptoms correlate with disease duration, suggesting that both may improve as patients adapt to chronic disability [76]. Addressing mental health concerns and promoting the development of skills through personal or therapy engaging strategies may help to overcome limitations thereby improving mood and quality of life in LHON patients. 

### 6.3. Induced Pluripotent Stem Cells

As stated, the chronic phase has particularly challenging limitations in treatment. Once optic atrophy has occurred there is currently no avenue to reverse this process or regrow damaged fibers. The intention of the previously discussed therapies was maximizing remaining vision and prevention of further loss. Induced pluripotent stem cells (iPSCs) present a potential remarkable and spectacular avenue for reversal of vision loss. This field of research could in theory be effective for such a broad scope of etiologies of vision that it would revolutionize the field. At this time, however, it is an extremely challenging option with many hurdles to overcome before becoming clinically feasible. Unlike other cellular derived lines which are more amenable to stem cell related therapy, the neuronal pathway a RCG must travel the re-establish cerebral connections is exceedingly difficult. 

In 2006, technology to reprogram somatic cells from patients to human iPSCs was developed [77]. Since then, development of this methodology has been revolutionized to allow for differentiation of iPSCs into RCGs [78,79,80]. Even if this technology does not evolve into an intervenable clinical therapeutic option, it serves as excellent mechanism of in vitro research. In relation to LHON differentiated RGCs and mini eye organoids have been created as disease models for study [81]. A few studies have been able to identify ways to stimulate RGC axons part way through the optic nerve and even beyond the chiasm. Mechanisms by which this has been accomplished include enhancement of mTOR, augmentation of adenosine 3′,5′-monophosphate (cAMP), and injections of oncomodulin [82]. Additionally, there is exciting advancement in the field of electrical field application to promote RGC survival and direct axon regeneration [83]. For iPSCs to be effective they would need to be appropriately differentiated, safely delivered to the right retinal location, and have the ability to form connections with the other retinal cells. Then, even if this can be accomplished, their axons would have to traverse to the optic nerve, decussate at the chiasm, and travel to the LGN to establish appropriate synaptic connections. Currently, the use of iPSCs is not yet feasible for vision loss related to optic nerve damage; however, the field has shown tremendous growth. The ultimate goal of this method would be to determine how to combine iPSC protocols with molecular pathway activation and RGC support to allow for their survival and regrowth. 

### 6.4. Mitochondrial Gene Editing

Given the limitations in current therapeutics in patients with LHON, the concept of mitochondrial genetic engineering presents a novel and exciting focus. Gene editing tools such as zinc-finger nucleases (mitoZFN) and transcription activator-like effector nucleases (MitoTALENs) have been shown to eliminate mutant mtDNA and shift heteroplasmy [84]. Recent advancements within the field have led to a new approach, mitochondrial base editing. An appropriate mitochondrial targeted DNA base editor might allow for the direct revision of the mutation of interest to wildtype mtDNA. One such example, TALE-linked adenine deaminases (TALEDs), has been tested in human cell lines and was able to edit A-to-G conversion at different target sites within the mitochondrial genome [85,86]. Although still far from clinical use, improvements in these methodologies are making it increasing possible to create targeted therapy against pathogenic mtDNA mutations. This innovative approach adds another potential field of interest in LHON therapy. 

## 7. Conclusions

In conclusion, the field of therapeutic options for LHON is still limited but has recently shown a dramatic expansion in scientific research and potential. This article reviewed supportive, preventative, and interventional therapeutics, how they are currently applied to clinical disease stages, and future promising areas within the field. 

The novel insights in molecular mechanism and cellular pathology found in LHON has created multiple new approaches for a deeper understanding of LHON and other mitochondrial optic neuropathies and related diseases. Presently, Idebenone remains the cornerstone of treatment and the only currently licensed therapy; however, the vast arsenal of potential therapies spans everything from repurposing established drugs, creation of novel molecules and gene therapy. The future of the field holds the promise of more efficacious treatment options for the treatment of LHON optic neuropathy. 

## Figures and Tables

**Figure 1 ijms-23-13205-f001:**
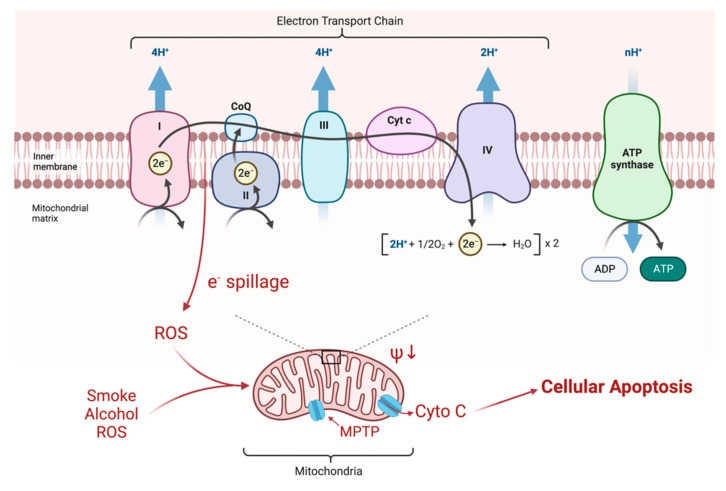
Electrons are transferred along the chain of complexes for oxidative phosphorylation. The energy is largely stored as a proton gradient that is later exploited for the production of ATP. Mutations of Complex I, as seen in LHON, produce a minor impairment of ATP production (about 20%), but also a greater production of ROS (10X) by spillage of electrons at the attempted transfer to Co Enzyme Q10. These ROS change the electrical potential across the mitochondrial membrane which, in crossing a critical threshold, can open the mitochondrial permeability transition pore (MPTP) releasing cytochrome C (Cyto C) and initiating apoptosis of the RGC [26].

**Figure 2 ijms-23-13205-f002:**
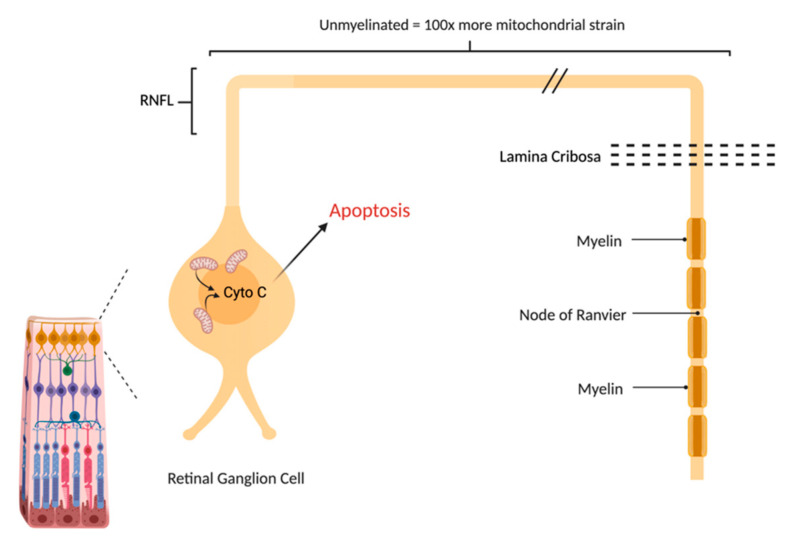
The Retinal Ganglion Cell (RGC) is particularly vulnerable to mitochondrial impairment. Like other neurons, it consumes a great deal of ATP in keeping its membrane potential, that is depleted after every axon potential. Unlike other neurons, it does not fully benefit from the economy of myelin that restricts the membrane changes to the Nodes of Ranvier. The long retinal nerve fiber layer (RNFL) is unmyelinated due to the need for retinal optical transparency, creating a great deal of extra metabolic strain on RGCs.

**Figure 3 ijms-23-13205-f003:**
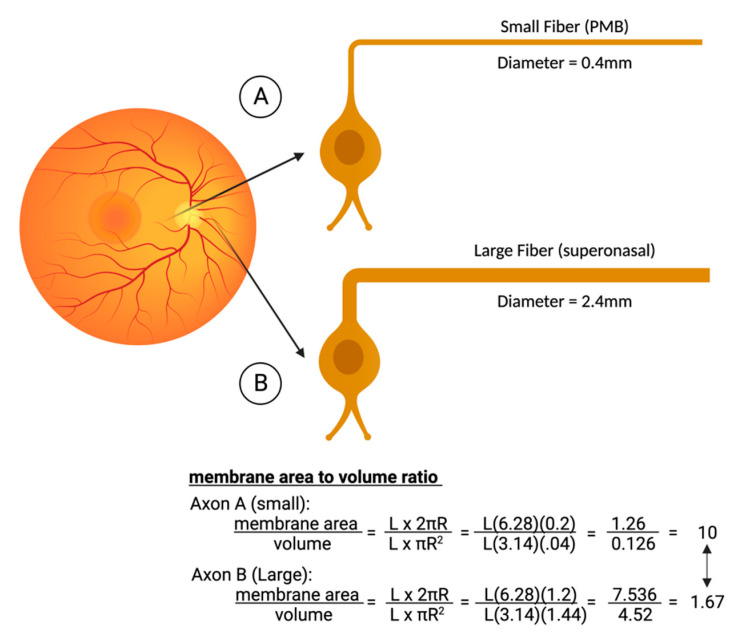
Comparison of the metabolic strain in smaller axons that are concentrated in the papillomacular bundle (PMB). The smaller fiber (**A**) typically has a diameter of only 0.4 microns. Compared to the larger fibers of the nasal retina (**B**), the PMB fiber has a much-reduced volume and hence fewer mitochondria. Although its surface area is a little less as well, the area/volume ratios are six times worse in the smaller fibers, explaining why they are the first to die in LHON. Figures created with biorender.com.

**Table 1 ijms-23-13205-t001:** Disease Classification.

Disease Classification
Asymptomatic (mutation carriers)
Preclinical
Subacute (<6 months)
Dynamic (6–12 months)
Chronic (>12 months)

## Data Availability

Not applicable.

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
