# Peer review of "Solutions to a Radical Problem: Overview of Current and Future Treatment Strategies in Leber’s Hereditary Opic Neuropathy"

_ijms, 2022, doi:10.3390/ijms232113205_

Round 1

Reviewer 1 Report

The present paper reports a well-documented and up-to-date analysis of the molecular basis, clinical time course and treatment possibility of LHON. The presentation is clear and all the present and future options to cope with the disease deeply discussed. Minor point in line 147 “in the” is repeated twice.

Author Response

Thank you for the review. It is much appreciated. The adjustment for typo in line 147 has been made. 

Reviewer 2 Report

Thank you for presenting the excellent paper. It is a great pleasure for me to read this very interesting review.

I think the manuscript covers a wide range of topics from pathogenesis to treatment of LHON.

I would like the authors to consider the following points ;

â‘ While many studies have been reported on idebenone showing its therapeutic effect, there are some papers that examine the pathogenesis of cases in which idebenone is not effective (Varricchio,C et al. Free Radic Biol Med, 2020, DOI:10.1016/j.freeradbiomed.2019.11.030). If possible, I would like to add to the discussion.

â‘¡Electrical stimulation has also been tried as a treatment for LHON (Kurimoto, T. et al.  J Clin Med, 2020, DOI: 10.3390/jcm9051359) (Ueda, K. et al. BMJ Open, 2021, DOI: 10.1136/bmjopen-2021-048814). Trans-corneal or trans-skin electrical stimulation has also been done for other optic nerve or retinal degenerative diseases, and its efficacy has been reported. I would like to have this added to the discussion, if possible.

Other minor points are; 

1. Line104 m.14484TCA -> m.14484T>C?

2. Line119 sube-acute→sub-acute?

3. Line321 14488→14484?

Author Response

The authors would like to thank the reviewer for their comments and appreciate the additions suggested. 

1) The authors agree and have added this to the discussion section of Idebenone under section 5.3.

2) The authors agree and have added this to the discussion under section 5.7 and broadened the category from light therapy to included electrical stimulation.  

Minor points as listed below have been adjusted. Thank you very much for picking up on those. 

1. Line104 m.14484TCA -> m.14484T>C?

2. Line119 sube-acute→sub-acute?

3. Line321 14488→14484?

Reviewer 3 Report

The manuscript by Spiegel and Sadun is a nice and useful review that provides an overview of LHON, covering both the clinical, pathophysiological and therapy aspects.

I have a few comments.

Comments:

·         Line 134: Only for the m.3460 mutation a clear impairment of Complex I enzymatic activity was reported, although the final effect for all three mutation is a defect in mitochondrial respiration and reduced Complex I-driven ATP synthesis. It is possible that the m.11778 and m.14484 mutations affect the proton pumping or coupling of electron transfer and proton pumping. I think that this point should be clarified.

·         Line 245: I think that the authors should mention also autophagy in this paragraph, as counterpart of mitochondrial biogenesis, not only transmitophagy occurring in the RGCs.

·         Line 398: the authors should explain the concept of allotopic expression, for which wild type ND4 is recoded based on the nuclear DNA genetic code to allow its translation in the cytoplasm.

·        As expanding fields, in addition to iPSCs and retinal organoids, I suggest to briefly introduce the new techniques for mtDNA editing (base editors), which may represent novel gene therapy approaches aimed at correcting the LHON mutations (Barrera-Paez and Moraes, Trends in Genetics 2022).

Minor comments:

Line 104: “14484TCA” must be corrected in “14484T>C”

Line 129: the authors probably missed the number of Complex I subunits (45)

Line 142: “there” must be corrected in “the”

Line 389: there is a typing error (mtDN instead mtDNA)

Author Response

Major Points: 

"Line 134: Only for the m.3460 mutation a clear impairment of Complex I enzymatic activity was reported, ...."

The authors agree and have adjusted this section to appropriate reflect this. Thank you for noting this important distinction. 

"Line 245: I think that the authors should mention also autophagy..."

The authors agree with this and have added to this section. We have also adjusted the title of this section to reflect this as well.

"Line 398:  the authors should explain the concept of allotopic expression, for which wild type ND4 is recoded based on the nuclear DNA genetic code to allow its translation in the cytoplasm..."

Thank you for this comment. We have added this explanation to section 5.8 "gene therapy" as requested. 

"As expanding fields, in addition to iPSCs and retinal organoids, I suggest to briefly introduce the new techniques for mtDNA editing (base editors), which may represent novel gene therapy approaches aimed at correcting the LHON mutations (Barrera-Paez and Moraes, Trends in Genetics 2022)."

Thank you for this suggestion as well. Being less familiar with this topic it required some additional reading, but hope that it addresses the reviewers comments appropriately. We have added an additional section, '6.4 mitochondrial gene editing'. 

Minor Points: Thank you for picking up on theses. They have been adjusted accordingly.